Development of a cognitive bias methodology for measuring low mood in chimpanzees

Bateson Melissa melissa.bateson@ncl.ac.uk
Nettle Daniel
Centre for Behaviour and Evolution and Institute of Neuroscience, Newcastle University , Newcastle upon Tyne , UK
Yuan Tifei
Electronic publication date: 2015 Jun 11
Publication date: 2015
Volume: 3
Electronic Location ID: e998
Received 2015 Mar 11; Accepted 2015 May 14
Copyright: © 2015 Bateson and Nettle
Copyright year: 2015
Copyright holder: Bateson and Nettle
License: This is an open access article distributed under the terms of the Creative Commons Attribution License, which permits unrestricted use, distribution, reproduction and adaptation in any medium and for any purpose provided that it is properly attributed. For attribution, the original author(s), title, publication source (PeerJ) and either DOI or URL of the article must be cited.
License URL: https://creativecommons.org/licenses/by/4.0/

Keywords: Cognitive bias, Judgment bias, Affective state, Chimpanzee, Depression

Funding: Centre for Behaviour and Evolution at Newcastle University NC3Rs NC/K000802/1 BBSRC BB/J016446/1 Field work was funded by a small grant from the Centre for Behaviour and Evolution at Newcastle University. MB was additionally supported by BBSRC grant BB/J016446/1 and NC3Rs grant NC/K000802/1, and DN was supported by BBSRC grant BB/J016446/1. The funders had no role in study design, data collection and analysis, decision to publish, or preparation of the manuscript.

==============================
There is an ethical and scientific need for objective, well-validated measures of low mood in captive chimpanzees. We describe the development of a novel cognitive task designed to measure ‘pessimistic’ bias in judgments of expectation of reward, a cognitive marker of low mood previously validated in a wide range of species, and report training and test data from three common chimpanzees (Pan troglodytes). The chimpanzees were trained on an arbitrary visual discrimination in which lifting a pale grey paper cone was associated with reinforcement with a peanut, whereas lifting a dark grey cone was associated with no reward. The discrimination was trained by sequentially presenting the two cone types until significant differences in latency to touch the cone types emerged, and was confirmed by simultaneously presenting both cone types in choice trials. Subjects were subsequently tested on their latency to touch unrewarded cones of three intermediate shades of grey not previously seen. Pessimism was indicated by the similarity between the latency to touch intermediate cones and the latency to touch the trained, unreinforced, dark grey cones. Three subjects completed training and testing, two adult males and one adult female. All subjects learnt the discrimination (107–240 trials), and retained it during five sessions of testing. There was no evidence that latencies to lift intermediate cones increased over testing, as would have occurred if subjects learnt that these were never rewarded, suggesting that the task could be used for repeated testing of individual animals. There was a significant difference between subjects in their relative latencies to touch intermediate cones (pessimism index) that emerged following the second test session, and was not changed by the addition of further data. The most dominant male subject was least pessimistic, and the female most pessimistic. We argue that the task has the potential to be used to assess longitudinal changes in sub-clinical levels of low mood in chimpanzees, however further work with a larger sample of animals is required to validate this claim.

Introduction

Objective, well-validated methods for assessing the presence of negative moods in great apes are currently lacking (Brüne et al., 2006). There are both ethical and scientific reasons why the development of such measures would be important in these species. First, the welfare of captive great apes is a matter of great public concern, and it is the legal and ethical duty of care-givers to identify and alleviate suffering in these animals. The presence of negative affective states such as anxiety and depression is a likely source of suffering and poor welfare, and even less severe low mood within the normal range is, by definition, unpleasant (Nesse, 2000). However, without the tools to correctly diagnose negative mood states, it is difficult to treat them effectively. Second, from a scientific perspective, affective state could be an important source of uncontrolled variation, or even a confound, in experiments on great apes (Rosati et al., 2013). For these reasons, the aim of the current study was to develop a novel, behavioural measure of depression and low mood in chimpanzees applicable to animals living in laboratory, zoo or sanctuary environments.

There have been some previous attempts to develop validated measures of mood in chimpanzees. King & Landau (2003) developed a questionnaire-based instrument designed to measure a chimpanzee equivalent of the human trait of subjective well-being (SWB). Zoo keepers were required to rate animals in their care on four scales relating to: pleasure derived from social interactions, balance of positive and negative moods, success in goal attainment and the desirability of being a particular chimpanzee. This methodology produced a single factor that had high inter-rater reliability and that correlated well with the objective measures of submissive behaviour. More recently, Ferdowsian and colleagues (2011, 2012) developed criteria for diagnosing post-traumatic stress disorder, major depression and anxiety disorder in chimpanzees based on modification of the criteria for these disorders in humans listed in DSM-IV (the Diagnostic and Statistical Manual of Mental Disorders, 4th edition). Necessary criteria for depression included at least one of, “Depressed hunched posture, social withdrawal, or easily irritated or angered”, or, “Loss of interest in food, play, other individuals or grooming” (Ferdowsian et al., 2011). As in the previous study, chimpanzees were assessed by trained individuals who were familiar with the animals concerned. The results suggested that 59% of captive chimpanzees showed symptoms of major depression compared with only 3% of wild chimpanzees; similarly, 18% of captive chimpanzees showed symptoms of generalized anxiety disorder, compared with only 0.5% of wild animals.

A major concern with the above methods is the use of descriptors that require the rater to make an assessment of the subjective state of a non-verbal, non-human animal using simple similarity to the expression of moods in humans. Despite the close phylogenetic relationship between humans and chimpanzees, chimpanzee behaviour is very different from that of humans, leading to the possibility of incorrect inferences about their underlying moods (Rosati et al., 2013). To avoid such anthropomorphism, it would be better to use a measure of mood that can be assessed by independent observers not familiar with the animals in question, using objective criteria that are derived from an understanding of the evolutionary function of moods. Furthermore, it would be useful to have a measure of low mood capable of detecting more subtle, sub-clinical mood states that would not meet the criteria for clinical anxiety or depression.

The strongest current candidate for such a measure is the assessment of so-called cognitive biases (Mendl, Burman & Paul, 2010; Mendl et al., 2009). Mood is an integrative function of an individual’s acute emotional experiences over time (Mendl, Burman & Paul, 2010; Nettle & Bateson, 2012), and can be defined operationally as a relatively enduring affective state that arises when negative or positive experiences in one time period alter the individual’s threshold for responding to potentially negative or positive events in subsequent time periods (Nettle & Bateson, 2012). This definition implies that mood can be assessed objectively by measuring these latter thresholds. Response thresholds are typically manifested in biases in attention to, memory of or judgment of potentially positive or negative events. Thus, for example, an animal that has suffered a series of social defeats might attend less to social information, might constantly recall previous defeats and might judge itself to have a poor chance of success in any future competition. Over the past decade, methodologies have been developed to assess such cognitive biases as objective measures of mood (Mendl et al., 2009).

The literature exploring the use of cognitive biases to assess mood in non-human animals has focused on the measurement of judgment biases when animals are asked to interpret ambiguous information. In a typical judgment bias task (e.g., Bateson & Matheson, 2007), subjects are initially trained to associate one stimulus—positive—with a high-valued reward and another stimulus—negative—with either punishment or lack of reward. Once the subjects have acquired the discrimination between the positive and negative stimuli, they are subsequently tested by presenting them with ambiguous stimuli intermediate between the two trained stimuli. In the test trials, animals that respond to the ambiguous stimuli similarly to the positive stimulus are interpreted as displaying a high expectation of reward in the presence of ambiguous information, and hence an ‘optimistic’ cognitive style indicative of a positive affective state. In contrast, animals that respond to the ambiguous stimuli similarly to the negative stimulus are interpreted as displaying a higher expectation of punishment or lower expectation of reward, and hence a more ‘pessimistic’ cognitive style indicative of a more negative affective state. Such cognitive bias tasks are currently regarded as the gold standard for assessing moods in non-human animals because they test clear, a priori predictions about the relationship between cognition and affective state that emerge from a general definition of moods and that are central to their evolutionary function (Mendl et al., 2009).

Manipulations hypothesized to produce changes in mood have been shown to be associated with the predicted shifts in judgment bias in a wide range of species from honeybees to humans (Anderson et al., 2012; Bateson et al., 2011). For example, in the first study of this type, Harding and colleagues (2004) trained rats trained on an auditory discrimination in which they had to press a lever following a positive 2 kHz tone to obtain a food reward but refrain from pressing the lever following a negative 4 kHz tone to avoid a blast of white noise. They showed that rats subjected to a chronic mild stress manipulation previously argued to induce a depression-like state were both slower and less likely to press the lever following a near-positive but ambiguous 2.5 kHz tone not previously heard than a control group not subjected to the manipulation. This judgment bias was present in the absence of any anhedonia (measured via a standard sucrose consumption test), suggesting that the task was sensitive to subtle, sub-clinical variation in mood state.

Although methodologies for measuring judgment bias have subsequently been developed for two monkey species (Bethell et al., 2012; Pomerantz et al., 2012), so far there have been no attempts to develop a judgment bias task for any great ape species. Our aim was therefore to develop a judgment bias task to assess expectation of reward, and hence depression-like low mood, in common chimpanzees (Pan troglodytes) living in a sanctuary environment. Our specific objectives were as follows: first, to develop a simple judgment bias task requiring only cheap, low-tech equipment that could be implemented in a range of chimpanzee housing facilities, including sanctuaries in developing countries; second, to develop an effective and efficient protocol for training the initial discrimination between positive and negative stimuli; and third, to establish the number of test sessions necessary to obtain stable measures of cognitive bias in individual chimpanzees and the number of test sessions possible before chimpanzees learn that the intermediate stimuli are never reinforced and the task therefore ceases to work. This paper describes the development of the task and presents detailed data from three animals living at Chimfunshi Wildlife Orphanage, Zambia, in service of the specific objectives described above.

We chose to develop a go/no-go judgment bias task based on similar tasks for European starlings (Sturnus vulgaris) previously used in Bateson’s lab (Bateson & Matheson, 2007; Bateson et al., Unpublished data). In such tasks, the subject is trained to make an active, ‘go’ response (in this case lift a paper cone) when presented with a positive stimulus (a pale grey cone), but withhold this response, ‘no-go’, when presented with a negative stimulus (a dark grey cone). The alternative to a go/no-go task is a ‘choice’ task, in which the subject has two possible active responses available. Choice tasks have been argued to be preferable to go/no-go tasks because they allow a general lack of motivation to respond to be distinguished from a pessimistic interpretation of an ambiguous stimulus (Matheson, Asher & Bateson, 2008; Enkel et al., 2010). Our rationale for choosing a go/no-go task in preference to a choice task was that previous experience suggested that go/no-go tasks are faster to train. For example, starlings took an average of 32.5 trials to learn the simple discrimination necessary for a go/no-go task (Bateson et al., Unpublished data) but an average of 238.5 trials to learn the conditional discrimination necessary for a choice task (Brilot, Asher & Bateson, 2010). We used the same achromatic visual discrimination used previously with starlings because, like starlings, chimpanzees are visually oriented animals, and achromatic colour discriminations have previously been acquired successfully by great apes (Schrauf & Call, 2009). Positive stimuli were rewarded with peanuts, and negative stimuli were associated with no reward. We chose not to associate negative stimuli with a punisher (such as toxic mealworms, white noise or electric shocks) because we were reliant on chimpanzees volunteering for our study, and therefore did not want to do anything that might deter them. An implication of this decision is that the judgment bias task only provides information about the chimpanzees’ expectations of reward and not about their expectations of punishment. Since biases in expectation of reward are typically associated with depression and biases in expectation of punishment with anxiety, the task provides information about depression-like negative moods but not about anxiety-like negative moods (Mendl et al., 2009; Nettle & Bateson, 2012).

Methods

Ethical statement

Approval for the project was obtained from the Chimfunshi Research Advisory Board and the Animal Welfare and Ethical Review Body of Newcastle University (ID number 390). All chimpanzees participated voluntarily in the project. The protocol required them to enter a room in which they were isolated and confined during testing (<1 h), but they were released from the testing room immediately if they showed signs of distress or were unmotivated to participate in the experiment (i.e., they did not come to the testing window for the pre-trial peanut when called—see below). No food deprivation was necessary, and we relied instead on using peanuts—a high value food that the Chimfunshi chimpanzees were usually strongly motivated to obtain.

Subjects

Subjects were three adult common chimpanzees, two males (Bobby aged 20.5 and Nicky aged 22) and one female (ET aged 20), located at Chimfunshi Wildlife Orphanage Trust, Zambia. All animals had access to extensive outdoor enclosures but were used to entering buildings daily for feeding. Chimpanzees were invited inside for individual cognitive testing outside their normal feeding time (1130–1330).

Experimental set-up and apparatus

For testing, chimpanzees were individually admitted to a room equipped with a window onto a corridor with a concrete table on either side of the window, such that the chimpanzee could sit on the table in the room and interact with stimuli placed on the table outside the room by the experimenter. The window was fitted with metal bars and we additionally attached a piece of 2.5-cm steel mesh that completely covered the window with the exception of a rectangular hole 30 cm wide × 10 cm high located in the centre, bottom of the window. The purpose of the mesh was to increase experimenter safety by restricting the area in which the chimpanzee could reach between the bars of the window.

The stimuli used in the experiment were paper cones constructed from 9-cm diameter paper circles, cut to the centre and fastened into a cone with a small piece of transparent sticky tape. The paper was standard white printer paper printed with a grey scale to indicate the valence of the stimulus. We used 20% grey for the positive (henceforth POS) stimulus and 60% grey for the negative (NEG) stimulus for all subjects. Intermediates were near positive (NP) in 30% grey, mid (M) in 40% grey and near negative (NN) in 50% grey. In POS trials the cone concealed a single, shelled, redskin peanut, whereas in both NEG trials and intermediate test trials there was nothing beneath the cone.

Cones were presented on a rectangular, red, plastic tray approximately 50 × 30 cm. Latencies were recorded live with a stopwatch, and test trials were filmed using a video camera on a tripod located in the corridor.

General procedure

For all trials the procedure was as follows. The chimpanzee was admitted to the experimental room. At the beginning of each session and following each break in a session the experimenter (MB) shook a box of peanuts in front of the window and called the chimpanzee in order to ascertain whether the chimpanzee was motivated to proceed. If the chimpanzee approached the window and accepted a peanut then trials began. The tray was prepared out of sight of the chimp. Depending on the phase of the experiment (for details see below) the tray was prepared with either a single cone placed on one side of the tray (forced trials), or two cones (choice trials), one POS and one NEG, placed on opposite sides of the tray approximately 45 cm apart; the POS cone always concealed a single peanut hidden underneath that the chimpanzee could obtain by lifting up the cone. The experimenter approached the concrete table and placed the tray on the table with the long edge nearest to the chimpanzee approximately 30 cm from the bars, such that the chimpanzee could pull the tray towards itself to reach the cone(s). The trial was timed from the moment the tray was put down on the table (an audible cue) until the chimpanzee first touched a cone (the latency to respond). If the chimpanzee did not touch a cone within 60 s of the start of the trial the tray was removed and the inter-trial interval (ITI) of 30 s was started. If the chimpanzee touched a cone, the ITI was started, the chimpanzee was allowed to take the peanut if one was present and the tray was removed as soon as it was safe to do so (i.e., when the chimpanzee had withdrawn his/her hands). Sessions comprised 24 trials during training and 25 trials during testing (with the exception of some pilot training sessions; see Table 2 for details) and always contained at least one break (see details below). We were opportunistic in our choice of subjects, and sometimes ran more than one session per half day if an animal was motivated. If there were multiple sessions within the same half day (0800–1100 or 1500–1700) for the same individual, there was a break of at least 5 min between sessions.

The experiment had several phases of training and testing. These are summarized in Table 1 and described in detail in the following sections. Figure 1 depicts the different types of trial in each phase of the experiment.

Figure 1 Methods summary.

Diagram depicting the different types of trial in each phase of the experiment.

Table 1 Summary of number of session in each of the various phases of the training and testing protocol.

Phase	Pilot	Block training (forced trials)	Choice	Test phase	
Sub-phase		6-trial blocks	3-trial blocks	Choice 0	Choice 1	Test 1	Choice 2	Test 2	Choice 3	Test 3	Choice 4	Test 4	Choice 5	Test 5	
Number of sessions	Variable	Minimum of 3. Continue until cumulative latencies are significantly different for three successive sessions.	Minimum of 1. Continue until cumulative latencies are significantly different.	Minimum of 1. If significant preference proceed to Choice 1. If NS return to 3-trial blocks.	1	1	1	1	1	1	1	1	1	1	

Table 2 Summary of discrimination training for each chimpanzee.

Chimp.	Pilot training	Block training: 6-trial blocks	Block training: 3-trial blocks	Total training trialsa	First session of choices following block training (i.e., Choice 0)	
	No. sessions	No. trials	Sessions to criterion	Mean POS latency (s)	Mean NEG latency (s)	p-valueb	Sessions to criterion	Mean POS latency (s)	Mean NEG latency (s)	p-valueb		No. trials	Proportion of POS choices	p-valuec	
Bobby	1	21	6	3.66	6.64	0.012*	3	3.81	5.99	0.022*	237	24	0.83	0.002*	
ET	1	11	3	4.74	35.22	<0.001*	1	13.34	33.03	0.013*	107	24	0.92	<0.001*	
Nicky	2	48	7	3.25	7.13	0.005*	1	0.92	1.82	<0.001*	240	24	0.83	0.002*	
Means			5.33				1.67				194.67				
Notes.

a Total training trials is equal to the number of pilot trials plus the number of block training trials.

b p-values come from t-tests.

c p-values come from binomial tests.

* p < 0.05.

Pilot training

Pilot sessions were used to establish the methodology for presenting the options to the chimpanzees and measuring their performance. In these sessions the POS and NEG stimuli (paper cones) were identical to those used in the later stages of the experiment, but the precise manner of presentation varied between chimpanzees and between sessions. Parameters explored included: whether the cones were presented in forced trials or choice trials; for forced trials, whether trials of the same valence were clustered in blocks; the length of the inter-trial interval; the distance of the tray from the chimpanzee; the spatial arrangement of the cone(s) on the tray. All three chimpanzees received at least one session of pilot training. The number of sessions and trials of pilot training received by each chimpanzee is summarized in Table 2, and the pilot training trials are included in the ‘Total training trials’ column of Table 2, since these trials could have contributed to the chimpanzees’ acquisition of the discrimination. The outcome of the pilot training was the training protocol outlined in the General Procedure above and the sections below. This latter protocol was applied consistently for the three chimpanzees after they had completed their pilot training. Had we trained further chimpanzees on the task, no pilot training would have been necessary for these animals, and they would have started with block training (see below).

Block training: six-trial blocks

The block training trials were used to teach the chimpanzee the discrimination between the POS and NEG cones. In the initial stages of this discrimination training, trials were presented in blocks of six forced trials (i.e., only one cone presented in each trial) of the same valence, alternating blocks of six POS and six NEG trials. Thus sessions comprised two blocks of POS trials and two blocks of NEG trials for a total of 24 trials. The first session always started with a block of six POS trials, and thereafter we alternated which valence started each session. The side on which each cone was presented was chosen pseudorandomly such that cones appeared an equal number of times on the right- and left-hand sides in each block. Within a block the ITI was 30 s. Between blocks there was a break of at least 2.5 min.

Note that for Bobby and Nicky only, a small piece (approx. 1 cm cube) of carrot was placed on the opposite side of the tray from the cone during all block training trials. The rationale for introducing the carrot was to provide an alternative item of interest, but of lower value than a peanut, that might attract the chimpanzee away from the cone in NEG trials. However, since these two chimpanzees rarely touched the carrot before the cone (1.27% of all trials for Bobby and 6.25% for Nicky), the carrot was abandoned for ET. No carrot was present in the choice or test sessions for any chimpanzee.

In each trial the latency to touch the cone was recorded; if a chimpanzee did not respond within 60 s its latency for that trial was coded as 61 s. Following each session we used a two-tailed, two-sample t-test to test whether the latency to touch the cones in POS trials was significantly shorter than the latency in NEG trials using all of the data collected thus far in the current phase. The criterion for progression to the next phase of training was three successive block training sessions in which this cumulative test was significant at p < 0.05 in the predicted direction.

Block training: three-trial blocks

This phase of training was identical to the former with the exception that blocks were reduced to just three trials of the same valence and hence sessions comprised eight blocks, four of each valence in an alternating sequence. The side on which each cone was presented was chosen pseudorandomly such that POS and NEG cones appeared an equal number of times on the right- and left-hand sides within a session. The criterion for progression to the next phase of training (Choice) was one session in which the cumulative t-test was significant at p < 0.05 in the predicted direction.

Choice

The rationale for including choice sessions was to confirm that the chimpanzees placed a higher value on the POS than the NEG cones. Choice sessions comprised 24 choice trials in which one POS and one NEG cone were presented simultaneously on the right- and left-hand sides of the tray. The side on which the POS cone was presented was varied pseudorandomly such that POS and NEG both appeared an equal number of times on the right- and left-hand sides within a session. The ITI was 30 s, and there was a break of at least 2.5 min following the twelfth trial. In each trial, the latency to touch the first cone and valence of this cone was recorded. It was generally not possible to safely withdraw the tray immediately following the chimp’s first choice, as has been possible in previous chimpanzee discrimination learning experiments (e.g., Spence, 1936). Therefore, it was possible for chimpanzees to touch both cones and hence receive the peanut reward even if they touched the POS cone second.

Following the first choice session (Choice 0), the criterion for progression to the test phase of the experiment was a significant departure from chance on a binomial test (i.e., 18 or more choices for POS from a total of 24 trials). If chimpanzees met this criterion they progressed to the test phase (which started with another choice session—Choice 1). The three chimpanzees tested in the current paper all met this criterion on their first session of choice trials. Within the test phase of the experiment, each test session (see below) was preceded by a further session of choice trials (in addition to the choice session they needed to have passed in order to proceed to this phase—see Table 1). A chimpanzee was required to have a significant preference in each choice session in order to progress to the test session.

Test

Test sessions always followed 5 min after the successful completion of the required choice session (see above). Test sessions comprised 25 forced trials including: eight POS, eight NEG, and three each of NN, M, and NP. In test trials, the cone was always presented on the right-hand side of the tray in order to remove any variance in latency due to subtle side biases in performance. Trials were presented in a pseudorandom order (the same for all chimpanzees, but different for each session within a chimpanzee). All test sessions began with POS or NEG and the trials were arranged such that POS and NEG trials were approximately evenly distributed across the session with intermediate trials interspersed between them. The ITI was 30 s and there was a 5 min break after trial 13. There were five test sessions for each chimpanzee yielding 15 trials on each intermediate and 40 trials on POS and NEG in total. In each trial the latency to touch the cone was recorded.

Statistical analysis

The raw data on which the analyses presented in this paper are based are published with the paper as Supplemental Information 1. All statistical analysis was conducted in R version 3.0.3 (R Core Team, 2013); the script is available on request. Where necessary, dependent variables were transformed prior to analysis to homogenise the variance and correct the distribution of residuals. Latencies were transformed using the reciprocal transformation (i.e., 1/latency) and proportions of POS choices were square rooted. Note that the predictor variable “chimp” is specified as a random effect in models in which we are seeking to make general inferences about the behaviour of chimpanzees, but as a fixed effect when we seeking to make specific inferences about the three individual chimpanzees included in our study.

For general linear mixed models containing random effects we used the R package “nlme” (Pinheiro et al., 2014). Model estimation was by maximum likelihood, and whether parameters differed significantly from zero was determined by testing the change in deviance when a given predictor was excluded from the model using a X2 test.

Results

Discrimination training

The chimpanzees displayed no neophobia towards the tray or cones, and in all cases immediately reached for and picked up the cones presented to them on the tray. Thus, no special training was required for the chimpanzees to perform the basic task, and all three animals could have progressed immediately to block training.

The amount of training received by each chimpanzee is summarized in Table 2. Once the chimpanzees started the blocked training they took a mean of 5.33 sessions of six-trial blocks to reach criterion in this phase, followed by a further 1.67 sessions of three-trial blocks. The female chimpanzee, ET, completed her block training in the minimum possible number of block-training sessions (4). The mean total number of trials taken by the chimpanzees to learn the discrimination (i.e., pilot trials plus block training trials) was 194.67 (range: 107–240).

All three chimpanzees showed a significant preference for touching the POS cone first in the session of choice trials immediately following successful completion of block training (Choice 0), demonstrating that a significant difference in latency in forced trials translated into a significant preference in choice trials (Table 2).

Discrimination performance during testing

The period of testing spanned 13 days for Bobby, 6 for ET and 12 for Nicky. Of note, both Bobby and Nicky had a gap of 6 days during which they were neither trained nor tested between test sessions 3 and 4. In the first set of analyses, we tested whether the chimpanzees retained the discrimination between POS and NEG learnt during training in the test phase of the experiment. Figure 2 summarises performance in the choice trials. All three chimpanzees showed a highly significant preference for touching the POS cone first in all choice sessions (binomial tests: all p < 0.01), with all three animals showing perfect performance in at least two of the five sessions. A general linear mixed model on the proportion of POS choices in a session (square-root transformed), with session number as a continuous predictor, and chimpanzee as a random effect, showed that the chimpanzees’ preference for the POS cone became stronger over the six choice sessions (effect of session: B ± se = 0.015 ± 0.004, X2(1) = 11.38, p < 0.001). Although the NEG cones never concealed peanuts, and the chimpanzees had clearly acquired the POS-NEG discrimination perfectly by the end of the testing, all three chimpanzees continued to pick up both cones on the majority of choice trials (data not shown).

Figure 2 Choice data.

Proportion of choices of the POS cone in each choice session for each of the three chimpanzees. The dotted line shows the criterion for choice to be significantly different from random (p < 0.05).

Figure 3 shows all the latencies from the five test sessions excluding trials on which the chimpanzees did not respond within 60 s. In the five sessions of test trials there was some variation in the chimpanzees’ motivation to respond and/or their attention to the task. Of the 125 test trials received by each animal, Bobby responded within 60 s in all 125 trials, Nicky in 123 and ET in 87 (numbers of trials for each test session are given in Fig. 3). The trials in which ET did not respond were clustered mainly in her first and last test sessions, and were not obviously higher in the intermediate than in the POS trials, as would be expected if she had learned that the intermediate cones were never reinforced (the proportions of each trial type in which she did not respond were: NEG = 0.38, NN = 0.33, M = 0.13, NP = 0.33 and POS = 0.28). In some of these trials she might have been reacting to the valence of the cone on offer, but in others she did not look at the cone during the entire 60 s of the trial, and her lack of response therefore had to be independent of the cone’s valence. For this reason we made the conservative decision to exclude all test trials in which a chimpanzee did not respond within 60 s from the analyses of the test trial data in order to remove some of the noise.

Figure 3 Test data.

Individual latencies to touch cones of each valence in each of the five test sessions for each of the three chimpanzees. Note that latency is plotted on a log axis to aid display of the data. The solid line in each panel shows the mean latencies for that test session. The number of trials (n) out of a maximum of 25 in each session in which the chimpanzee touched the cone within 60 s is given in each panel.

Individual two-sample t-tests showed that all three chimpanzees had significantly shorter latencies to touch POS cones than NEG cones during testing (Table 3; all p < 0.05). Thus, the chimpanzees’ performance on the choice and forced trials shows that the POS-NEG discrimination established during training was retained, and even strengthened, over the testing period.

Table 3 Results of t-tests comparing latencies to touch POS and NEG cones during testing.a

Chimpanzee	T statistic	df	p-value	
Bobby	−2.54	72.34	0.013*	
ET	−6.60	75.00	<0.001*	
Nicky	−2.45	51.41	0.018*	
Notes.

a These tests were performed on the POS and NEG latencies from all 5 test sessions.

* p < 0.05.

Figure 3 suggests that there were differences between chimpanzees, and also between test sessions within a chimpanzee, in their mean latency to touch POS and NEG cones, and hence their overall speed. A general linear model on the latencies to touch POS and NEG cones in the test sessions, with chimp, session and their interaction as categorical fixed predictors, but for now not considering the valence of the cones, showed that these latencies differed between chimpanzees and sessions (interaction of chimp × session: F(8, 197) = 2.55, p = 0.011). Due to these differences in mean speed, it was necessary to control statistically for a chimpanzee’s mean latency to touch POS and NEG cones in a test session in any analysis of its latency to touch the intermediate cones, by including speed as a covariate (see below).

Response to intermediate cones

Next we explored whether there were differences between chimpanzees in how they responded to the intermediate cones. A general linear model on the latencies to touch NP, M and NN cones, with chimp, valence (continuous) and their interaction as predictors and speed as a continuous covariate, showed that these latencies differed between chimpanzees (effect of chimp: F(2, 116) = 15.52, p < 0.001); there was also a near-significant interaction between chimpanzee and valence (interaction of chimp × valence: F(2, 116) = 2.54, p = 0.083). Together, these results show that the three chimpanzees responded differently to the intermediate cones, even when their overall speed to touch POS and NEG cones was controlled for. A Tukey HSD test on the effect of chimpanzee showed that Nicky had significantly lower latencies to touch intermediate cones than both Bobby and ET, but that Bobby and ET were not significantly different from each other (Table 4).

Table 4 Results of Tukey HSD test showing which chimpanzees showed significantly different responses to the intermediate cones after correction for multiple testing.

Chimpanzee pair	Difference in observed meansa	Lower end point of 95% CI	Upper end point of 95% CI	Adjusted p-value	
ET-Bobby	−0.03	−0.22	0.16	0.938	
Nicky-Bobby	0.33	0.16	0.51	<0.001*	
Nicky-ET	0.36	0.17	0.55	<0.001*	
Notes.

a Due to the use of reciprocal latency in the model, these values are reversed in magnitude. Hence, the true ranking in mean latency to intermediates is: Nicky < Bobby < ET.

* p < 0.05.

To make the differences between chimpanzees easier to visualize, the mean latencies from the five test sessions for each chimpanzee are shown in Fig. 4A. Nicky responded to all of the intermediate cones similarly to the POS cone, Bobby also responded to NN and NP similarly to POS (his response to M was very variable), whereas ET responded to intermediate cones at a speed intermediate to POS and NEG. These differences can be more easily seen in Fig. 4B in which the intermediate latencies for each chimpanzee are standardized by expressing them as a proportion of the difference between its latencies to touch the POS and NEG cones. To obtain a single number that provides an index of pessimism for each chimpanzee, we computed each individual’s mean latency to touch all three intermediate cones and again expressed this as a proportion of the difference between its latencies to touch the POS and NEG cones: pessimism index = (mean intermediate latency − mean POS latency)/(mean NEG latency − mean POS latency). This index is equal zero if a chimpanzee responds to the intermediate cones at the same speed as to the POS cones, and to one if it responds to the intermediate cones at the same speed as to the NEG cones (note that negative pessimism indices are possible if mean POS latency >mean NEG latency, and values greater than 1 are possible if mean intermediate latency is >mean NEG latency; however these situations should rarely arise if an animal has learnt the task and mean POS latency <mean intermediate latency <mean NEG latency). The pessimism indices for the three chimpanzees computed using all the data from the five test sessions are shown in Fig. 4C. ET emerges as the most pessimistic chimp, Bobby as the next most pessimistic and Nicky as the least pessimistic.

Figure 4 Summary test data.

(A) The mean latencies (±1 sem) in the five test sessions for each chimpanzee to touch cones of each valence. (B) The same data shown in (A) standardized such that the POS latency for each chimpanzee is equal to 0 and the NEG latency equal to 1. (C). The pessimism index for each of the three chimpanzees (see text for details).

Effect of number of test sessions

To explore whether, and if so how the number of test sessions affected the results obtained, we explored how the results changed as we added more test session data. Figure 5 shows how three key statistics, namely, the mean speed to touch POS and NEG cones, the effect size of the difference in latency to touch POS and NEG cones (expressed as Cohen’s d, i.e., the mean difference divided by the pooled sd) and the pessimism index (defined above) changed with the number of test sessions. The data are plotted in two ways. The first plot for each statistic (left-hand column of Fig. 5) shows the results for each individual test session, whereas the second plot (right-hand column of Fig. 5) shows the cumulative results computed using all of the test data collected up to the current session. Although the individual session statistics are quite variable from session to session, the cumulative statistics show a clearer pattern, as would be expected. First, there are consistent individual differences between the three chimpanzees in mean speed: the rank order of the chimpanzees is preserved across the five test sessions, and there is no evidence for a consistent change in speed. Second, the effect size of the POS-NEG discrimination is consistently around 0.5 (generally considered as a medium-sized effect). Finally, from the second test session onwards, there are consistent individual differences in the pessimism index: the rank order of the chimpanzees is preserved across the remaining four test sessions, and importantly there is no evidence that chimpanzees are becoming more pessimistic with time, as would be expected if they were learning that the intermediate cones were never reinforced.

Figure 5 Analysis of the effects of including progressively more sessions of test data on the test results.

The graph shows three key statistics summarizing the chimpanzees’ performance in the test sessions. (A) and (B) mean speed to touch POS and NEG cones. (C) and (D) Cohen’s d for the differences in latency to touch POS and NEG cones—a measure of effect size. (E) and (F) pessimism index (see text for definition). (A), (C) and (E) shows these statistics in each test session for each chimpanzee, whereas (B), (D) and (F) shows the cumulative versions of the same data presented in the left-hand column.

Discussion

The aim of this study was to develop a low-tech cognitive bias task for assessing depression-like low mood in chimpanzees that could be applied in a sanctuary environment. We developed a judgment-bias task inspired by similar tasks used successfully in other animal species. These tasks provide a measure of the subject’s expectation of reward in the face of ambiguous information, and hence an objective, behavioural measure of a cognitive trait analogous to the pessimism typical of human low mood and depression (Mendl et al., 2009). We trained chimpanzees that a positive stimulus—a pale grey paper cone—was associated with a peanut reward, whereas a negative stimulus—a dark grey paper cone—was associated with no reward, and demonstrated that they could acquire this arbitrary visual discrimination. We subsequently tested the chimpanzees by measuring their latencies to respond to ambiguous cones intermediate in shade between the two trained shades. We succeeded in training and testing three chimpanzees on this task at Chimfunshi wildlife orphanage. Although this number of animals is small, we were able to collect enough data to suggest that this task may provide meaningful and stable measures of individual differences in pessimism, and hence depression-like low mood. Below we discuss some methodological issues relating to the development of the task and some ideas for how it could be used in future research.

Task development and limitations

The task that we designed required only cheap, low-tech equipment (a tray and printed paper), and only minimal modification of the chimpanzees’ cage (temporary attachment of the safety mesh to the barred window). It was possible to implement the task in a sanctuary without dedicated research facilities and lacking resources such as mains electricity. Opting for a go/no-go task meant that both discrimination training and judgment bias testing could be achieved with forced trials, in which only a single stimulus (cone) was presented in each trial. One advantage of this approach was that it avoided the need for custom-built apparatus that prevented the chimpanzee taking both options, as would have been possible if animals were presented with a choice of two stimuli (e.g., Spence, 1936 used a pair of lockable metal boxes mounted on a track to present choices to his chimpanzees and permit only a single selection to be made).

Despite never being reinforced on the NEG trials, the chimpanzees that we tested rarely withheld touching the NEG cone within 60 s. For this reason, we were obliged to use latency to touch the cones as our dependent measure as opposed to a binary go/no-go response (c.f. Bateson & Matheson, 2007). An advantage of using latencies is that they are a continuous, and therefore potentially more sensitive, measure of what the subject has learnt. However, latencies are also particularly vulnerable to variation in motivation and external disturbance as is clearly evident in Fig. 3. Due to this variability, it was often not possible to detect significant differences in latency to touch POS and NEG cones within a single session, and for this reason we introduced sessions of choice trials (see Table 1) as a means of assessing quickly whether the chimpanzees had acquired/retained the discrimination.

We addressed the potential problem of differences in motivation to respond inherent in go/no-go tasks (Matheson, Asher & Bateson, 2008; Enkel et al., 2010) by controlling for motivation in our statistical analyses. To do this, we included the mean latency to touch the POS and NEG cones in a given session as a covariate in our analysis of the latency to touch the three intermediate cones in that session (the same procedure was adopted in Bateson et al., unpublished data). The pessimism index (Fig. 3) also controls for motivation by expressing the latency to touch the intermediate cones as a proportion of the difference between the latency to touch the POS and NEG cones. Using these two approaches, we obtained estimates of pessimism that should be independent of the overall speed of a chimpanzee to touch cones on a given day, and thus insensitive to day-to-day fluctuations in motivation. Despite considerable day-to-day variation in the chimpanzees’ latencies evident in Fig. 3, our statistical analysis revealed a significant difference between chimpanzees in their latencies to touch intermediate cones, with the female chimpanzee ET and the male chimpanzee Bobby emerging as significantly more pessimistic than the male chimpanzee Nicky (Table 4). We interpret this result as showing individual differences in pessimism, and hence low mood, that are not simply explained by individual differences in motivation to do the task.

Our protocol relies on animals voluntarily repeatedly entering the test room to obtain peanuts during the task. This requirement potentially imposed a bias in the animals that we were able to test, because the extreme low mood found in clinical depression is associated with anhedonia, fatigue and social withdrawal, all of which might militate against participation in a food-motivated task. Therefore, whilst our task might be suitable for measuring subtle individual variation in sub-clinical levels of low mood, it might miss animals with severe depression that are unwilling to participate or not interested in treats. It is worth noting that this criticism applies to most of the judgment bias studies on non-human animals published to date, since the majority have used food as the reward associated with the positive stimulus (Mendl et al., 2009).

Training the discrimination

A major challenge for the current project was to develop an efficient protocol for training the initial visual discrimination between pale grey cones and dark grey cones necessary for the judgment bias task. Many published judgment bias tasks required extensive discrimination training that would not be feasible in settings such as Chimfunshi, where it is difficult to test specific individuals on a regular schedule. Thus, minimizing the number of training sessions required was a priority. We were concerned that the recent literature states that great apes find it difficult to learn arbitrary visual discriminations. For example, Schrauf & Call (2009) found that only 6/12 apes (bonobos, gorillas and orangutans) learnt an arbitrary achromatic colour discrimination (black versus white) in 36 trials of training, leading them to conclude that, “learning to associate an arbitrary cue such as colour or weight with a reward is not a trivial task for great apes”. Similarly, Hanus & Call (2011) found that chimpanzees failed to learn an arbitrary colour discrimination (again black versus white), but interestingly could learn a discrimination where there was a causal relationship between the stimuli and reward, in 15 trials of training.

We succeeded in training three adult chimpanzees on an arbitrary visual discrimination using blocks of forced trials and latency to touch the stimulus as the outcome measure. Our rationale for training with forced trials as opposed to choices arose from the fact that it was impossible for us to withdraw the tray immediately following the chimpanzee’s first choice, meaning that chimpanzees typically took both cones on a choice trial. Given this constraint, we reasoned that it would be difficult for chimpanzees to associate choosing a pale grey cone with reward, since on choice trials they typically took both cones and always got rewarded. Training with forced trials avoided this problem.

Our rationale for using blocks of trials of the same type arose from the fact the literature on associative learning shows that animals of all species learn associations faster when inter-trial intervals are longer relative to the conditioned stimulus (CS) exposure time, because the CS (here the cone colour) provides better information about the arrival of the unconditioned stimulus (the peanut) (Gallistel & Gibbon, 2000; Balsam & Gallistel, 2009; Ward, Gallistel & Balsam, 2013). However, when we tried to use long inter-trial intervals to speed up training, the chimpanzees became frustrated. The solution was to use a short inter-trial interval, which proved critical for maintaining the chimpanzee’s interest in the task, whilst simultaneously reducing interference between the different trial types and preserving a clear association between the cone presented and reward by presenting blocks of multiple trials of the same type separated by longer intervals. We started with blocks of six trials and subsequently reduced this to blocks of three trials to dissuade the chimpanzees from simply learning a win-stay strategy within the current block.

Using this training procedure our chimpanzees took between 107 and 240 trials to reach criterion on the discrimination task. Interestingly, whilst this is a far greater number of trials than the maxima tried in the recent great ape studies reported above, it is a similar number to those reported in older literature on visual discrimination learning in great apes (Spence, 1936; Rumbaugh & Rice, 1962; Davis, 1978). Using a procedure similar to ours, Spence (1936) succeeded in training 12 adult chimpanzees on a series of arbitrary visual discriminations with the animals taking 101.25 + 48.3 (mean + sd; range 40–220) trials to reach criterion. From his data, Spence concluded that chimpanzees learn arbitrary visual discriminations using similar mechanisms, and at a similar speed, to other vertebrate species tested. Our results concur with this conclusion. It is of note that our previous attempts to train starlings on simple visual discriminations have resulted in much faster training (see Introduction for figures). The main differences between our starling experiments (Bateson et al., Unpublished data; Bateson & Matheson, 2007) and the current chimpanzee experiment are, first, the inclusion of punishment (in the form of bitter quinine) associated with NEG stimuli in the starling experiments, and second much longer inter-trial intervals (4–5 mins) in the starling experiments. It would be interesting to explore whether discrimination learning in apes could be speeded up via the introduction of either of these changes.

It is possible that our criterion for moving from training with blocks of six trials to blocks of three trials was overly conservative, since all the animals we trained showed a significant preference in their first session of choice trials (Choice 0). However, we were reluctant to intersperse earlier sessions of choice trials in the training as a more continuous indicator of progress (as was done by Brilot, Asher & Bateson, 2010), because we were worried that this might interfere with acquisition of the discrimination due to our inability to prevent the chimpanzees from taking both cones (see above).

Cognitive bias testing

The judgment bias task that we used relies on the intermediate stimuli being truly ambiguous in the test trials. To improve ambiguity, we used POS and NEG stimuli that were more similar than in some of our previous studies (here 20% grey versus 60% grey compared with 0% versus 80% in Bateson & Matheson, 2007). However, given sufficient training, the chimpanzees could potentially have learnt that the intermediate stimuli were different from POS and NEG and were never reinforced. This problem has been reported in previous judgment bias experiments (Doyle et al., 2010; Brilot, Asher & Bateson, 2010) and serves to restrict the number of test sessions that it is possible to conduct with an animal, and hence the usefulness of the task for assessing within-individual changes in affective state. One of our aims in the current study was therefore to collect several sessions of test data. This would allow us to establish, first, how quickly the chimpanzees learnt that the intermediate cones were never reinforced, and second, how estimates of pessimism were altered by the addition of additional test data. Both of these pieces of information would be useful for the design of future studies using the task.

We ran a total of five test sessions for each chimpanzee, and explored how the results changed as we added more test session data (Fig. 5). These data allowed us to establish the following information. First, the data show that individual differences in the pessimism index emerged in the second test session and that the rank order of these differences was not changed by the addition of three further sessions of test data. Thus, testing could potentially have been successfully achieved with as few as two test sessions. Second, there is no evidence that the chimpanzees became more pessimistic with successive test sessions, as would be expected if they were learning that the intermediate cones were never reinforced (for examples where this did occur see: Brilot, Asher & Bateson, 2010; Doyle et al., 2010). The failure of the chimpanzees to learn that the intermediate cones were never reinforced over five sessions of testing was perhaps due to the relatively difficult discrimination (20% versus 60% grey) used in the current experiment. Thus, it is possible that there is a trade-off between the difficulty of the initial positive-negative discrimination and the number of test trials it is possible to obtain from a judgment bias task. Third, although both Bobby and Nicky had a gap in testing of six days between test sessions 3 and 4, there is no evidence from Figs. 2, 3 or 5 indicating that the performance of these two chimpanzees declined significantly during this period. Together these data show that the performance of the chimpanzees on our task stabilized rapidly and then remained stable over time, even if there were gaps in training/testing of several days. These findings suggest that it would potentially be viable to use the task to assess longitudinal changes in affective state within individual chimpanzees over a period of at least a week. Further work would be needed to establish if chimpanzees could retain this task and discrimination over longer periods of time necessary for longer-term longitudinal studies.

We were successful in obtaining measures of pessimism from three chimpanzees. According to our data, Nicky (an adult male) was least pessimistic, Bobby (another adult male) was in the middle and ET (an adult female) was the most pessmistic. We interpret these differences in pessimism as suggesting that ET had the lowest mood of the three animals and Nicky the most positive. However, whilst it can be used to rank the mood of the chimpanzees, on its own our task says nothing about the severity of their symptoms. Further validation, ideally with antidepressant drugs, would be needed to establish whether the levels of pessimism we measured in ET were indicative of clinically significant depression (e.g., Anderson, Munafò & Robinson, 2013).

During the period of our study we observed that Nicky appeared more gregarious than Bobby, and was frequently seen playing with juvenile males, whereas ET preferred to spend time on her own. ET had also lost her only daughter within the past year, leaving her with no genetic relations within her group. For these three animals there was also a correlation between pessimism and the relative ranks reported in Nettle, Cronin & Bateson (2013), with the highest ranking chimpanzee (Nicky) being the least pessimistic and the lowest ranking chimpanzee (ET) being the most pessimistic. We did not re-measure dominance in the current study, and thus it is possible that the relative ranks of Nicky and Bobby could have changed since 2013. However, since female chimpanzees always rank below adult males, the relative ranking of ET within the trio we studied is sound. King & Landau (2003) reported a negative correlation between SWB, as derived from their questionnaire-based instrument, and an objective measure of the number of submissive behaviours performed by a chimpanzee, also suggesting a relationship between dominance and SWB. With a sample of three (two males and one female) it is impossible to draw any conclusions from these anecdotal observations about the possible associations between sex, rank and low mood, but it would be interesting to follow this up in future research. In support of our observations, there is sample evidence in the human literature for associations between low mood/depression and sex, low social status, social isolation and recent bereavement. That these associations should be found across species makes sense in the light of thinking about the evolutionary functions of low mood (Nesse, 2000; Keller & Nesse, 2005).

In conclusion, we have developed a simple judgment bias task for chimpanzees designed to measure individual differences in pessimism, that requires only cheap, low-tech equipment and that can be implemented in a sanctuary environment. We have shown that it is possible to successfully train three adult chimpanzees on this task within as few as 107 trials, using a latency-based measure. The task produced stable individual differences in pessimism about reward. We argue that the task has the potential to be used to assess longitudinal changes in sub-clinical levels of depression-like low mood in chimpanzees, however further work with a larger sample of animals is required to validate this claim.

Supplemental Information

Supplemental Information 1 READ ME file giving details of the data file

Click here for additional data file.

Supplemental Information 2 Data file

Click here for additional data file.

We thank Katherine Cronin and the Chimfunshi Research Advisory Board for permission to do research at Chimfunshi; Innocent Mulenga and his staff for support at Chimfunshi; Gema Martin-Ordas for useful discussions and help collecting data.

Additional Information and Declarations

Competing Interests

Author Contributions

Animal Ethics

Field Study Permissions

The authors declare there are no competing interests.

Melissa Bateson conceived and designed the experiments, performed the experiments, analyzed the data, wrote the paper, prepared figures and/or tables, reviewed drafts of the paper.

Daniel Nettle conceived and designed the experiments, performed the experiments, analyzed the data, reviewed drafts of the paper.

The following information was supplied relating to ethical approvals (i.e., approving body and any reference numbers):

Approval for the project was obtained from the Chimfunshi Research Advisory Board and the Animal Welfare and Ethical Review Body of Newcastle University (ID number 390).

The following information was supplied relating to field study approvals (i.e., approving body and any reference numbers):

Approval for the project was obtained from the Chimfunshi Research Advisory Board.

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
