# Peer review of "Development of a cognitive bias methodology for measuring low mood in chimpanzees"

_PeerJ, doi:10.7717/peerj.998_

## Round 0.1 · original submission · Major Revisions

please revise the paper according to the comments of the reviewers.

Reviewer 1 ·

Basic reporting

This study described a cognitive task to measure the pessimism in chimpanzees, which can reflect the depression-like mood to some extent. However, we are very concerned about the findings with a sample of only three animals (two males and one female), and the authors should explain in more details why the cognitive task can measure depression-like mood. A possible correlation has been reported between social dominance and pessimism with the highest ranking male being the least pessimistic and the lowest ranking female being the most pessimistic, but the relevant data was not presented. Although the ranks have been reported in 2013, they may change over time. In order to exactly describe the relationship between them, the authors should re-evaluate the animals’ ranks. And we question whether the task performance was influenced by sex with the female took the longest time to respond to the stimuli.

Experimental design

We are so curious why the three chimpanzees received different number of sessions in pilot training, and we want to know whether it affects the animals’ performance in cognitive test.

Validity of the findings

In Table 2, the female (ET) was found to take less training trials to reach the criterion. Can this show the ET (female) is smarter than the other two chimpanzees (males) or it is only sex-specific? In Figure 3, we found the ET took the longest time to respond to the stimuli, but she took part in the least trials (87) than the other two chimpanzees (125 and 123). Although the ET is slower in responding to the stimuli, she learns very fast as exhibited by the least testing trials. Or she is so slow as to take more than 60 s to touch the cone (these trials were not included), but the data was not presented.

Additional comments

This is indeed a novel cognitive task to measure the pessimism in chimpanzees, but the findings are disputable. There are only three animals (one female and two males) performing the task, therefore, the validity of this methodology should be further confirmed when extended to a large sample. Furthermore, sex differences in the task performance should also be considered because it may have a great influence on the performance.

Reviewer 2 ·

Basic reporting

See below.

Experimental design

See below.

Validity of the findings

See below.

Additional comments

In this study, the authors developed a new paradigm to evaluate the depression-like behaviour in chimpanzees. Generally, this study was well conducted with solid and sound design. Also the figures are easy to be read, which makes the readers easier to go through this study. Therefore, I think this study meet the publishing criteria in PeerJ.

·

Basic reporting

No comments

Experimental design

The choice of the subjects is somehow biased because based on motivation (i.e. desire to get a peanut). Considering that depressed individuals show a lack of motivation, it is unlikely that the depressed-like apes would come and be tested… therefore it seems necessary to develop a way to test also the animals that did respond to the peanut cue and that are more likely to actually present a depressive-like profile.
This inclusion issue does not cast doubt on the results of this preliminary study but should be mentioned.

Validity of the findings

The goals are relevant and interesting in the current context of developing translational methods, directly transposable to Humans. The results suggest that most aims have been achieved. However, the methodology could be further adjusted to improve the reliability of the task (inclusion criteria) and further investigations are needed to confirm the results in a larger number of animals (n=3 here) and over time (line 609 the authors suggest the longitudinal potential of the task; the same animals would need to perform the task a couple of weeks or months later to assess that point).
Finally it’s important to state in the conclusion that although depressive patients display a judgement bias, this criterion only is not sufficient to diagnose depression. Therefore, though very interesting, this task only cannot guarantee that high pessimistic apes are indeed depressed.

Additional comments

This study aimed at developing a judgement bias task to assess expectation at reward in chimpanzees as a depression-like mood indicator. The goals were threefold: i) develop a cheap task that requires low-tech equipment; ii) develop an effective training protocol to discriminate positive and negative stimuli; and iii) establish the number of test sessions necessary to obtain stable measures of cognitive bias in individual chimpanzees and the number of test sessions possible before chimpanzees learn that the intermediate stimuli are never reinforced.

The manuscript is well-written (see below for the few mistakes to correct). The goals are relevant and interesting in the current context of developing translational methods, directly transposable to Humans. The results suggest that most aims have been achieved. However, the methodology could be further adjusted to improve the reliability of the task and further investigations are needed to confirm the results in a larger number of animals and over time.
I support the publication of this paper providing some corrections and specifications (see below for detailed comments and questions).

Methods
- The choice of the subjects is somehow biased because based on motivation (i.e. desire to get a peanut). Considering that depressed individuals show a lack of motivation, it is unlikely that the depressed-like apes would come and be tested… therefore it seems necessary to develop a way to test also the animals that did respond to the peanut cue and that are more likely to actually present a depressive-like profile.
However, this inclusion issue does not cast doubt on the results of this preliminary study.
- Lines 221 and 223: there are extra “s” to chimpanzees.
- Table 1: the font should be modified to increase readability.
- Table 2: what does “prop” stand for? Proportion?
- Line 297: there is a mistake for the number of intermediate cones. It is 3 instead of 2 according to the figure 1 and to line 305.
- Line 298: “in test trials” has been written twice in the same sentence.

Results
- ET has been tested on a smaller number of days and presented a smaller number of training trials. ET also showed the highest pessimism index.
Could it be possible that the index was inversely correlated to the number of trials? Three animals are not enough to perform reliable correlations. But could the following hypothesis work? The index is based on latencies to reach the cones. Even though an animal has reach criterion to discriminate positive and negative cones, it is likely that the speed increases until it reaches its maximum and remains stationary. ET might not have reached the maximum during the training sessions whereas Nicky and Bobby have. This may explain the differences in Figure 5.

- Line 362: ET responded within 60s in lesser trials than the other apes. The authors associated this with a lack of motivation. I argue that it may be associated with attention as well. Considering ET is at a lower social rank, one would expect that her gaze profile was different from the dominant apes, ie she would display more fast glances (to check if dominants come to get the peanuts and she should flight) than long stares. A dominant does not need to check for other apes’ approaches and can focus its attention on the task. Or does ET have a baby outside of the testing room?
The fact that these slow responses are independent of the valence of the stimuli corroborates the “attention” possibility.

- The figure 5 is a bit confusing and needs more detailed explanation.
Third row: pessimism index. The index is a number between 0 and 1 and there are 5 sessions. How can there be negative data or data > 5 on these graphs? I am not sure I understand correctly…

Discussion
- Lines 469-470: there are tracked modifications in blue.

- To avoid the ape from touching both cones on each test trial, would it be possible that the experimenter gives a reward to the chimp when it touches the POS cone and only this one instead of hiding the peanut under the cone? This might increase the association between the POS cone and the reward.

- Despite the very promising results, it’s important to state in the conclusion that a larger number of apes is required to confirm the outcomes of the task.
Moreover, (line 609) the same animals would need to perform the task a couple of weeks or months later to assess the “longitudinal” potential of the task.
Finally, although depressive patients display a judgement bias, the readers should not forget that this criterion only is not sufficient to diagnose depression. Therefore, though very interesting, this task only cannot guarantee that high pessimistic apes are indeed depressed.

---

## Round 0.2 · accepted · Accept

Congratulations on your great work!

Tifei

Reviewer 1 ·

Basic reporting

No Comments

Experimental design

No Comments

Validity of the findings

No Comments

Additional comments

No Comments

·

Basic reporting

The revised version is much clearer.
The authors have answer all of my questions and comments.

Experimental design

No further comments

Validity of the findings

No further comments

Additional comments

No further comments